# Cohort profile: a prospective Australian cohort study of women's reproductive characteristics and risk of chronic disease from menarche to premenopause (M-PreM)

Hsiu-Wen Chan ,[1] Shyamali Dharmage,[2] Annette Dobson,[1] Hsin-Fang Chung,[1] Deborah Loxton,[3] Jenny Doust,[1] Grant Montgomery,[4] Emmanuel Stamatakis,[5] Rachel R Huxley,[6,7] Mark Hamer,[8] Jason Abbott ,[9,10] Bu Beng Yeap,[11,12] Jenny A Visser,[13] Harold McIntyre,[14,15] Gregore Iven Mielke,[1] Gita D Mishra[1]

For numbered affiliations see end of article.

**Correspondence to**
Dr Gita D Mishra;
G.Mishra@uq.edu.au

## ABSTRACT

**Purpose** Previous studies have identified associations between individual reproductive factors and chronic disease risk among postmenopausal women. However, few have investigated the association of different markers of reproductive function, their interactions and risk factors of chronic disease among women approaching menopause. The Menarche-to-PreMenopause (M-PreM) Study aims to examine the relationship between reproductive factors across the reproductive lifespan and risk indicators for chronic disease among women in their early-to-mid-40s. The purpose of this cohort profile paper is to describe the rationale, study design and participant characteristics of the M-PreM Study.

**Participants** Women born in 1973–1978 who participated in the Australian Longitudinal Study on Women's Health (ALSWH) were invited to undertake a clinical or self-administered assessment. A total of 1278 women were recruited from June 2019 to June 2021.

**Findings to date** The study measures included functional, cognitive and cardiometabolic tests, anthropometry, spirometry, respiratory health questionnaires, physical activity, sleep patterns, sex hormones, and cardiovascular and metabolic markers; whereas blood and saliva samples were used for the analysis of genetic variants of genes associated with reproductive characteristics and chronic disease. The mean age of the clinic and self-assessed participants was 44.6 and 45.3 years, respectively. The menopausal status of participants was similar between the two arms of the study: 38%–41% premenopausal, 20% perimenopausal, and 36% took oral contraception or hormone replacement therapy. Approximately 80% of women had at least one child and participants reported experiencing pregnancy complications: preterm birth (8%–13% of pregnancies), gestational diabetes (10%) and gestational hypertension (10%–15%).

**Future plans** The biomedical data collected in the M-PreM Study will be linked to existing ALSWH survey data on sociodemographic factors, health behaviour, reproductive function, and early life factors collected over the past 20 years and health administrative data.

### STRENGTHS AND LIMITATIONS OF THIS STUDY

⇒ Biomedical data from the Menarche-to-PreMenopause Study will be linked to two decades of comprehensive survey data and administrative health service records.
⇒ A biobank of blood and saliva samples will enable the investigation of genetic factors that contribute to chronic disease risk.
⇒ The COVID-19 pandemic adversely impacted participation rates for this study and respiratory data collection.
⇒ Due to the complex blood processing requirements, we were only able to establish study sites in major cities.

The association between reproductive factors and risk indicators of chronic disease will be analysed.

## INTRODUCTION

Rapidly ageing populations in Australia and globally and the rising tide of associated chronic diseases present a pressing challenge for public health policy. Many age-related chronic diseases show marked sex differences in their prevalence and aetiology, with women being at a greater disadvantage overall. For example, although middle-aged women have a lower prevalence of diabetes,[1] women with diabetes are at greater risk of coronary heart disease,[2] stroke[3] and all-cause mortality[2] than men with diabetes. After adolescence, the prevalence of asthma is not only higher for Australian women than men across adulthood, but increases rapidly leading up to and through the menopausal transition.[4] Lastly, women have a higher prevalence of

non-amnestic mild cognitive impairment than men[5] and experience faster cognitive decline.[6]

There is a growing body of evidence on the links between reproductive factors and risk of chronic disease. Large pooled studies have shown that early menarche, early menopause and a short reproductive lifespan are associated with increased risk of cardiovascular disease in postmenopausal women.[7 8] Similarly, meta-analyses and systematic reviews demonstrate associations between early menarche, early menopause and increased odds of type 2 diabetes.[9 10] Early menarche is also linked to asthma in early-to-mid-adulthood.[11]

Few studies have investigated the associations between reproductive health with risk factors of chronic disease in younger women. Early menarche and longer menstrual cycle length (≥33 days) are associated with elevated diastolic blood pressure, total cholesterol, low-density lipoprotein (LDL) cholesterol, and triglyceride levels among women in their 30s and also those approaching menopause, which may be partially mediated by body composition.[12 13] Sex hormonal disorders in women of reproductive age, such as polycystic ovary syndrome (PCOS) and premature ovarian insufficiency, have also been linked to cardiometabolic risk factors and/or poor lung function and asthma. Women with PCOS not only have a higher body mass index (BMI)[14] and waist-to-hip ratio than other women, but they also have higher levels of total and free testosterone and dehydroepiandrosterone (DHEA).[15] In women with PCOS, there is some evidence that free testosterone is negatively associated with high-density lipoprotein (HDL) cholesterol levels, whereas DHEA is positively associated with triglyceride and LDL concentration.[16] Sex hormones are also associated with lung function. DHEA is negatively associated with forced expiratory volume in 1 s ($FEV_1$) and forced vital capacity (FVC) and with greater risk of airflow limitation later in life.[17] Other reproductive factors, including early menarche and parity, are associated with poorer lung function (lower $FEV_1$, FVC, and $FEV_1$/FVC; $FEV_1$/FVC is a marker of airway obstruction) and asthma.[11 18 19]

It has been long recognised that women are at higher risk of developing chronic diseases after menopause, especially cardiometabolic and respiratory disease. However, it is unclear whether reproductive history from adolescence and early-to-mid-adulthood affects the risk factors for chronic diseases in the years preceding menopause. Therefore, in order to implement targeted early preventative health strategies, there is a pressing need to examine the relationship between reproductive factors, sex hormones and risk factors of chronic disease in women before they enter the menopausal transition. Additionally, while previous studies have identified sex-specific risk factors for chronic conditions, many have viewed reproductive characteristics in isolation, rather than as a series of related and interacting factors.

The Menarche-to-PreMenopause (M-PreM) Study aims to:

1. Investigate links between reproductive factors and markers of cardiometabolic, respiratory, cognitive and functional health among women before they reach middle-age.
2. Determine whether body size (anthropometry) and physical activity modify these relationships.

The markers of cardiometabolic health measured are glycated haemoglobin (HbA1c), serum lipids and blood pressure, and the markers of respiratory health will be obtained from lung function tests and questionnaires. Other measures include functional and cognitive tests. In addition, this study will use existing data on physical and mental health, reproductive factors and sociodemographic data from the Australian Longitudinal Study on Women's Health (ALSWH) surveys and linked administrative health records. This paper describes the recruitment of participants, data collection protocol and participant characteristics of the M-PreM Study.

## COHORT DESCRIPTION

The ALSWH is a national, prospective study of women's biological and mental health, social and lifestyle factors, and health service use. In 1996, women born in 1921–1926, 1945–1951, and 1973–1978 were randomly selected from the database of Medicare (Australia's universal health insurance scheme) with intentional oversampling of women from rural and remote areas. Data collection of this currently ongoing study primarily comprises self-report surveys conducted approximately every 3 years. More information about the ALSWH can be found elsewhere.[20 21]

### Exclusion and inclusion criteria

The eighth survey of the 1973–1978 ALSWH cohort was conducted in 2018 when the participants were aged 40–45 years and completed by 7121 participants. In this survey, women were introduced to the M-PreM Study and asked to indicate their interest in participating. Those who responded positively (n=4584) were invited to the M-PreM Study (figure 1). The exclusion criteria included women who were pregnant or had been diagnosed with a breast or reproductive cancer and were undergoing treatment at the time of invitation. Chemotherapy and radiation treatments for breast or reproductive cancers often affect reproductive function and blood sex hormone concentration.[22]

### Sample size

The M-PreM Study commenced in June 2019 and concluded in June 2021. Due to the SARS-CoV-2 (COVID-19) outbreak in Australia, the study was suspended from March 2020 to November 2020. The operations of this study and participation have been adversely affected by sudden periods of lockdown in different Australian states, border closures, and new health and safety regulations for clinical research. Overall, 1278 participants were recruited: 499 participants had completed a

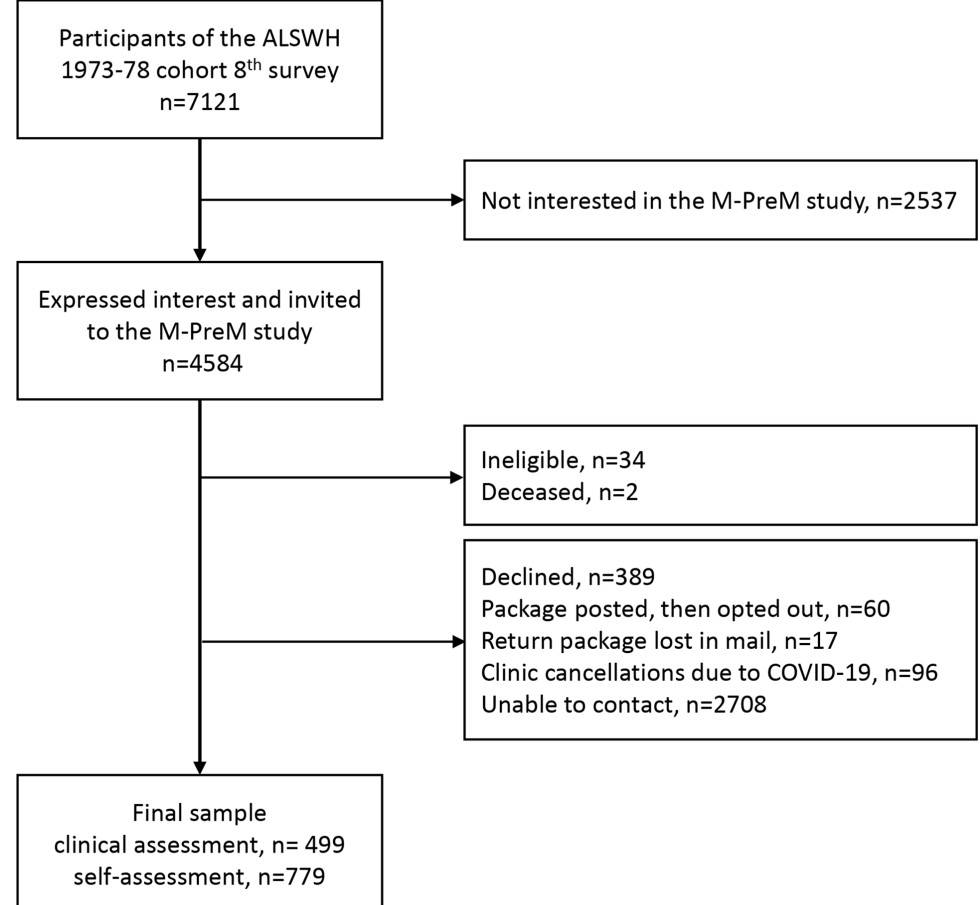

**Figure 1** The participant recruitment flow chart for the M-PreM Study. ALSWH, Australian Longitudinal Study on Women's Health; M-PreM, Menarche-to-PreMenopause.

clinical assessment and 779 participants completed a self-administered assessment (figure 1).

Based on conservative estimates, sample sizes of 499 and 779 for the clinical and self-administered assessments, respectively, provide enough power to detect associations between exposure–outcome (eg, the association between early menarche and blood pressure) of small to medium magnitudes after adjustment for up to 10 variables using linear regression models. Assuming a power of 80%, a significance level of 5% and a model including covariates that explains 10% of the variation in the outcome (reduced model; $R^2$ of 0.10), the minimum detectable values for the $R^2$ difference between the reduced and full models (model including 10 covariates and the exposure of interest) are 0.014 ($R^2$ of the full model: 0.114) and 0.009 ($R^2$ of the full model: 0.109) for the clinical and self-administered assessments, respectively. For a power of 90%, the minimum detectable values for the $R^2$ difference between the reduced and full models are 0.019 for the clinical assessment and 0.012 for the self-administered assessment.

### Data collection and measures

The M-PreM Study will analyse existing longitudinal and linked data from the ALSWH in combination with new biomedical data.

### Existing data from the ALSWH

The 1973–1978 cohort have completed up to nine surveys approximately every 3 years since 1996 (the ninth survey was conducted in 2021–2022; figure 2). From these surveys, data on sociodemographic factors, self-reported anthropometry, health-related behaviours and reproductive function are available. The specific items relevant for the M-PreM Study are detailed in table 1.

### Modes of participation

Study sites were established in the Translational Research Institute (Brisbane), Murdoch Children's Research Institute (Melbourne), Royal Hospital for Women (Sydney), PARC Clinical Research (Adelaide) and Harry Perkins Institute of Medical Research (Perth). Participants were invited to attend a clinical assessment where they undertook a range of biomedical measurements, provided a non-fasting blood sample, were fitted with a physical activity monitor which they wore for 8 days and completed an 8-day sleep diary (table 2). Participants who reported 'ever having PCOS' were also asked to complete a 3-month menstrual diary. Before the COVID-19 pandemic, participants conducted pre-bronchodilator and post-bronchodilator spirometry; however, this was replaced by a respiratory health questionnaire after the outbreak emerged for health and safety reasons (more

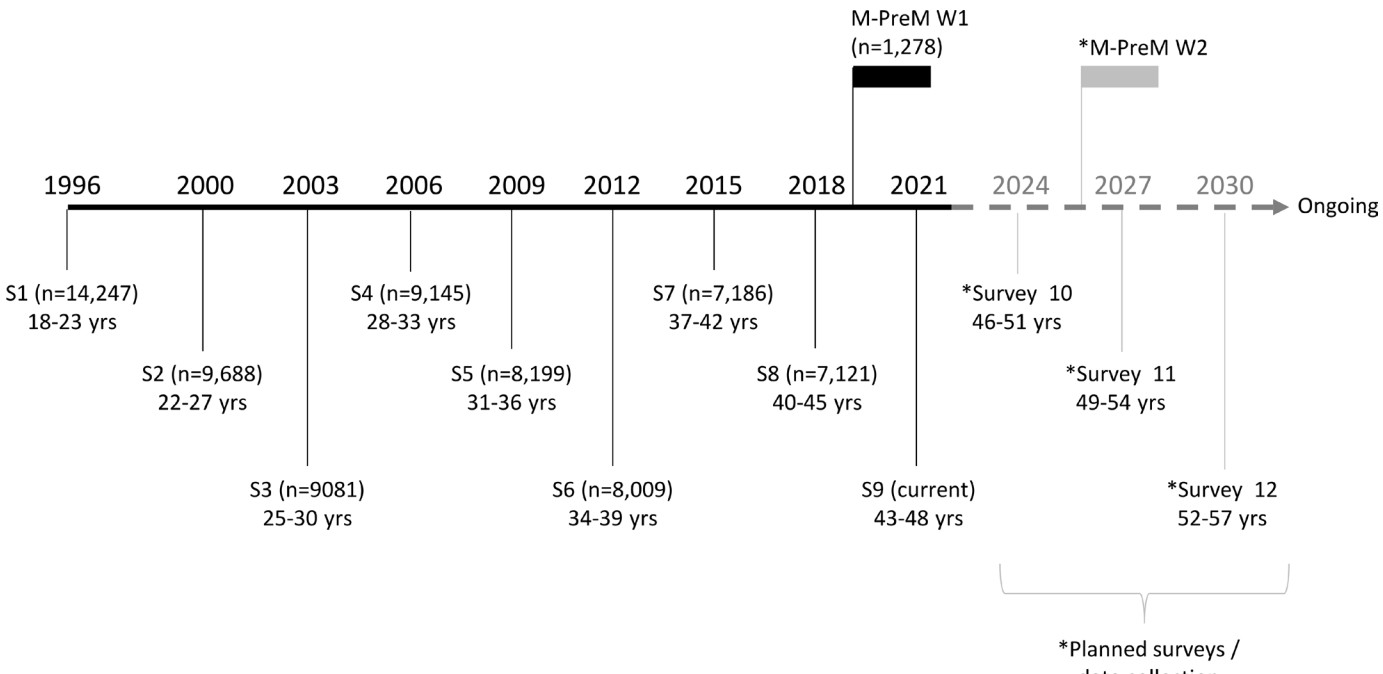

**Figure 2** Timeline and number of participants for the existing surveys (S1–9) and the M-PreM data collection of the ALSWH 1973–1978 cohort. The future surveys for this cohort are shown in grey. The M-PreM Study provides baseline (wave 1, W1) data for a potential follow-up (W2) of the participants. ALSWH, Australian Longitudinal Study on Women's Health; M-PreM, Menarche-to-PreMenopause.

details can be found in the Spirometry methods section). Where participants were unable to attend a study site, they were sent a package that allowed collection of self-administered data. This contained an Oragene·DNA saliva collection kit (OG-600; DNA Genotek) and an accelerometer (table 2). The package was then returned via Reply Paid post to the study team at the University of Newcastle.

## QUESTIONNAIRES

Reproductive characteristics, including use of contraception and hormone therapy, menopausal status and self-reported PCOS, have been captured using questions from the ALSWH surveys.[20] Menstrual cycle stage at the time of assessment was determined using questions from the Seattle Midlife Women's Health Study.[23] Respiratory health, including attacks of asthma or wheezy breathing, whistling in the chest, shortness of breath and chest tightness, was determined using the questionnaire from the Follow-up of the Tasmanian Longitudinal Health Study from first to sixth decade,[24] which was adapted from the European Community Respiratory Health Survey II.[25] Cognition was assessed using the Montreal Cognitive Assessment.[26]

## Handgrip strength test

Handgrip strength was measured (in kilograms) using Jamar hydraulic hand dynamometer (model 5030J1).[27] Participants were seated with a straight-back posture and their feet flat on the ground. They were asked to hold their elbow against their side at a 90° angle and their forearm and wrist in a neutral position. The handgrip strength test was performed once in each hand.

## Chair rise test

After removing their shoes, participants were asked to sit in a straight-backed armless chair with a horizontal, flat seat. Participants were assessed on the length of time (in seconds) taken to complete 10 sets of a sit–stand–sit manoeuvre.[28]

## Standing balance test

Participants were tested for the length of time (in seconds) that they could maintain a one-legged stance for a maximum of 30 s, first with their eyes open and then closed.[28] This test was performed without shoes and with both arms crossed over the chest. Participants were asked to stand on their preferred foot and to raise their non-preferred foot off the floor by bending their knee.

## Blood pressure and heart rate at rest

After resting in a seated position for 5 min, three measurements of systolic and diastolic blood pressures and heart rate, interspersed by 1–2 min breaks, were recorded using an automated blood pressure monitor. For analysis, only the second and third measurements were averaged, as recommended by the WHO protocol.[29]

## Anthropometric assessment

Weight without shoes was measured once using a digital scale to the nearest 0.1 kg. Height without shoes was measured once using a stadiometer to the nearest millimetre. Using these measurements, BMI ($kg/m^2$) was

**Table 1** Self-reported data from the ALSWH relevant to the M-PreM Study

| Category | Item* |
| --- | --- |
| Sociodemographic factors | Highest educational attainment<br>Area of residence<br>Marital status |
| Anthropometry (self-reported) | Weight<br>Height<br>Waist circumference |
| Health-related behaviour | Current and lifetime smoking history<br>Alcohol intake (quantity and frequency)<br>Use of illicit drugs<br>Level of physical activity<br>Sedentary behaviour, self-reported sitting time<br>Dietary intake, food frequency questionnaires |
| Reproductive function | Age at menarche<br>Dysmenorrhoea<br>Irregular periods<br>Premenstrual syndrome<br>Age at first delivery<br>Number of miscarriages<br>Terminations<br>Stillbirths<br>Live births<br>Diagnosis of endometriosis or polycystic ovary syndrome<br>Use of fertility treatment<br>Use of contraception<br>Hysterectomy and oophorectomy<br>Diagnosed with gestational diabetes or gestational hypertension<br>Premature birth<br>Infant birth weight<br>Duration of breast feeding |
| Early life factors | Early life nutrition (breast feeding) (adapted from the Longitudinal Study of Australian Children)<br>Mother's pregnancy complications<br>Attitude and behaviour of parents/caregivers during childhood[43]<br>Childhood household condition[44]<br>Childhood health[45] |

*Information for the standard questions administered in the ALSWH (not referenced in this table) can be found in 20 21.
ALSWH, Australian Longitudinal Study on Women's Health; M-PreM, Menarche-to-PreMenopause.

calculated. The waist circumference was measured at the midpoint between the bottom of the last palpable rib and top of the iliac crest (for self-assessment, this was described as 'the midpoint between bottom of the ribs and top of the hip'). The hip circumference was measured at the widest point of the buttocks (for self-assessment, this was described as the 'widest portion of the bottom'). The arm circumference was measured at the midpoint between the acromion process of the scapula and the olecranon process (for self-assessment, this was described as the

**Table 2** Questionnaires, measurements, biological samples collected and biological markers measured

| Category | Clinic | Self-administered |
| --- | :---: | :---: |
| Reproductive characteristics | | |
| Contraception use[20] | ✓ | ✓ |
| Hormone therapy use[20] | ✓ | ✓ |
| Menopausal status[20] | ✓ | ✓ |
| Ever had PCOS | ✓ | ✓ |
| Menstrual cycle staging[23] | ✓ | ✓ |
| 3-month menstrual diary (if ever had PCOS) | ✓ | ✓ |
| Respiratory health* (before 13 March 2020) | | |
| Attacks of asthma or wheezy breathing[24 25] | ✓ | ✓ |
| Use of medications that affect the airways | ✓ | |
| Recent respiratory illness | ✓ | |
| Spirometry | ✓ | |
| Respiratory health* (after 13 March 2020) | | |
| Attacks of asthma or wheezy breathing[24 25] | ✓ | ✓ |
| Attacks of cough[24 25] | ✓ | ✓ |
| Phlegm in the chest[24 25] | ✓ | ✓ |
| History of pneumonia and other chest illnesses[24 25] | ✓ | ✓ |
| Ever had bronchitis, COPD and emphysema | ✓ | ✓ |
| COVID-19 | ✓ | ✓ |
| Cognition | | |
| Montreal Cognitive Assessment[26] | ✓ | |
| Functional tests | | |
| Handgrip strength test[27] | ✓ | |
| Chair rise test[28] | ✓ | |
| Standing balance test[28] | ✓ | |
| Anthropometry | | |
| Height | ✓ | ✓ |
| Weight | ✓ | ✓ |
| Body mass index | ✓ | ✓ |
| Waist circumference | ✓ | ✓ |
| Hip circumference | ✓ | ✓ |
| Arm circumference | ✓ | ✓ |
| Thigh circumference | ✓ | ✓ |
| Physical activity | | |
| ActivPAL accelerometers worn for 8 days[46] | ✓ | ✓ |

Continued

**Table 2** Continued

| Category | Clinic | Self-administered |
|---|---|---|
| Sleep patterns | | |
| 8-day sleep diary[46] | ✓ | ✓ |
| Biological samples | | |
| Whole blood | ✓ | |
| Serum | ✓ | |
| Plasma | ✓ | |
| Buffy coat | ✓ | |
| Saliva | | ✓ |
| Cardiometabolic tests | | |
| Blood pressure | ✓ | |
| Resting heart rate | ✓ | |
| Total cholesterol | ✓ | |
| HDL cholesterol | ✓ | |
| LDL cholesterol | ✓ | |
| Glycated haemoglobin | ✓ | |
| Sex hormones | | |
| Oestradiol | ✓ | |
| Estrone | ✓ | |
| Testosterone | ✓ | |
| Dihydrotestosterone | ✓ | |
| Progesterone | ✓ | |
| Dehydroepiandrosterone | ✓ | |
| Anti-Mullerian hormone | ✓ | |
| Sex hormone-binding globulin | ✓ | |
| Follicle-stimulating hormone | ✓ | |
| Luteinising hormone | ✓ | |

*The respiratory health questionnaire and measurements changed after the onset of the COVID-19 pandemic.
COPD, chronic obstructive pulmonary disorder; HDL, high-density lipoprotein; LDL, low-density lipoprotein; PCOS, polycystic ovary syndrome.

'midpoint between the top of your shoulder to the point of your elbow'). Thigh circumference was measured at the midpoint between the lateral superior margin of the patella and the anterior superior iliac spine (for self-assessment, this was described as the 'midpoint between the top of your hip to the top of your knee'). Circumference measurements in the clinic were conducted to the nearest millimetre in duplicate, whereas at home, participants were asked to record the measurements in centimetres to 1 decimal point. Where the difference between the two measurements exceeded 5 mm, a third measurement was performed. The average of the two or three measurements was then calculated.

### Spirometry

Lung function testing was conducted from 9 July 2019 to 13 March 2020, after which it was removed from the protocol due to health and safety concerns as a result of the COVID-19 pandemic. Lung function testing using Easy-on-PC spirometers (NDD Medizintechnik, Switzerland) was performed according to the American Thoracic Society and European Respiratory Society guidelines[30] and based on the protocols of the Tasmanian Longitudinal Health Study.[24] Participants were required to perform a minimum of three acceptable trials for each pre-bronchodilator and post-bronchodilator test. Post-bronchodilator spirometry was performed 10–15 min after the bronchodilator, salbutamol (300 µg), was administered via a spacer.

To replace the lung function testing in the context of the COVID-19 pandemic, an extended respiratory health questionnaire was then given to all participants who were recruited after the M-PreM Study resumed on 10 September 2020. In addition to the original respiratory health questions (detailed in the Questionnaires section), the following questions were asked: ever had and age of diagnosis of bronchitis, chronic obstructive pulmonary disorder and emphysema; and history of bronchitis, phlegm in the chest, cough in the absence of a cold, cough with phlegm, and chest illnesses including pneumonia. Like the original respiratory health questions, these questions were also sourced from the Follow-up of the Tasmanian Longitudinal Health Study from first to sixth decade.[24] Lastly, questions about COVID-19 testing and results as well as self-reported changes to the participant's respiratory function were in the extended respiratory health questionnaire.

Participants who had already completed a clinical or self-administered assessment prior to the onset of the COVID-19 pandemic were also invited to complete the extended respiratory health questionnaire.

### Physical activity and sleep patterns

Accelerometers (activPAL3 micro and activPAL4 micro devices; PAL Technologies, UK) were attached to the middle of the anterior surface of the right thigh using adhesive dressing (3M Tegaderm) and worn continuously for 8 full days. Participants were asked to log their sleep patterns (bedtime, estimated time they fell asleep, waking time and time they got out of bed) and sleep quality (number of night wakings, how well they slept) for 8 days.

### Biological samples and biomarker testing

BD Vacutainer EDTA (for plasma and HbA1c testing), SST II (for serum and cholesterol testing) and plain tubes (for serum) were used to collect non-fasting venous blood. Whole blood, serum, plasma and buffy coat were also extracted. All samples are biobanked at –80°C at the Institute of Molecular Biosciences (Brisbane, Australia).

Serum estrone, oestradiol, testosterone, dihydrotestosterone, progesterone and DHEA concentrations were analysed using a validated liquid chromatography-mass

**Table 3** Characteristics of the M-PreM participants and the remainder of the ALSWH 1973–1978 cohort

| | M-PreM clinical assessment n=499 (7.01%) | M-PreM self-assessment n=779 (10.94%) | 1973–1978 ALSWH survey 8 cohort, not in M-PreM n=5843 (82.05%) | P value* |
|---|---|---|---|---|
| Area of residence, n (%) | | | | |
| Major city | 463 (93.16) | 313 (40.44) | 3248 (57.40) | <0.0001 |
| Inner regional | 26 (5.23) | 275 (35.53) | 1594 (28.17) | |
| Outer regional | 4 (0.80) | 164 (21.19) | 695 (12.28) | |
| Remote | 4 (0.80) | 22 (2.84) | 122 (2.16) | |
| Missing | 2 | 5 | 184 | |
| Highest qualification, n (%) | | | | |
| Less than year 12 or equivalent | 10 (2.00) | 39 (5.01) | 305 (5.55) | <0.0001 |
| Year 12 or equivalent | 24 (4.81) | 68 (8.73) | 529 (9.62) | |
| Trade/apprenticeship/certificate/diploma | 92 (18.44) | 229 (29.40) | 1586 (28.85) | |
| University | 373 (74.75) | 443 (56.87) | 3078 (55.98) | |
| Missing | 0 | 0 | 345 | |
| Marital status, n (%) | | | | |
| Never married | 58 (11.62) | 77 (9.88) | 636 (11.55) | 0.771 |
| Married/de facto | 390 (78.16) | 624 (80.10) | 4299 (78.08) | |
| Separated | 26 (5.21) | 33 (4.24) | 236 (4.29) | |
| Divorced | 22 (4.41) | 40 (5.13) | 309 (5.61) | |
| Widowed | 3 (0.60) | 5 (0.64) | 26 (0.47) | |
| Missing | 0 | 0 | 337 | |
| Employment status, n (%) | | | | |
| Not in labour force | 60 (12.02) | 103 (13.24) | 840 (14.89) | 0.028 |
| Employed full time | 225 (45.09) | 330 (42.42) | 2529 (44.82) | |
| Employed part time | 180 (36.07) | 274 (35.22) | 1752 (31.05) | |
| Employed casually | 34 (6.81) | 71 (9.13) | 521 (9.23) | |
| Missing | 0 | 1 | 201 | |
| Self-reported health, n (%) | | | | |
| Excellent/very good | 300 (60.12) | 477 (61.23) | 3130 (53.89) | 0.0002 |
| Good | 157 (31.46) | 220 (28.24) | 2003 (34.49) | |
| Fair/poor | 42 (8.42) | 82 (10.53) | 675 (11.62) | |
| Missing | 0 | 0 | 35 | |
| Smoking status, n (%) | | | | |
| Never smoked | 344 (68.94) | 482 (61.95) | 3428 (61.00) | 0.0002 |
| Ex-smoker | 125 (25.05) | 235 (30.21) | 1587 (28.24) | |
| Current smoker | 30 (6.01) | 61 (7.84) | 605 (10.77) | |
| Missing | 0 | 1 | 223 | |
| Alcohol use, n (%) | | | | |
| Non-drinker | 39 (7.82) | 64 (8.23) | 609 (10.82) | 0.014 |
| Low risk | 432 (86.57) | 663 (85.22) | 4598 (81.73) | |
| High risk | 28 (5.61) | 51 (6.56) | 419 (7.45) | |
| Missing | 0 | 1 | 217 | |
| Number of children, n (%) | | | | |
| No children | 105 (21.04) | 127 (16.30) | 1149 (19.66) | 0.005 |
| 1 child | 56 (11.22) | 96 (12.32) | 789 (13.50) | |

**Table 3** Continued

| | M-PreM clinical assessment n=499 (7.01%) | M-PreM self-assessment n=779 (10.94%) | 1973–1978 ALSWH survey 8 cohort, not in M-PreM n=5843 (82.05%) | P value* |
|---|---|---|---|---|
| 2 children | 229 (45.89) | 345 (44.29) | 2328 (39.84) | |
| 3 or more children | 109 (21.84) | 211 (27.09) | 1577 (26.99) | |
| Missing | 0 | 0 | 0 | |
| Age at menarche, n (%) | | | | |
| 11 years or younger | 52 (12.53) | 74 (11.23) | 551 (11.51) | 0.749 |
| 12 years | 115 (27.71) | 185 (28.07) | 1371 (28.63) | |
| 13 years | 124 (29.88) | 202 (30.65) | 1482 (30.95) | |
| 14 years | 80 (19.28) | 106 (16.08) | 813 (16.98) | |
| 15 years or older | 44 (10.60) | 92 (13.96) | 571 (11.93) | |
| Missing | 84 | 120 | 1055 | |
| History of preterm birth, n (%) | | | | |
| Yes | 33 (8.38) | 87 (13.43) | 668 (14.49) | 0.003 |
| No | 361 (91.62) | 561 (86.57) | 3943 (85.51) | |
| Missing | 105 | 131 | 1232 | |
| History of stillbirth, n (%) | | | | |
| Yes | 11 (2.79) | 7 (1.07) | 92 (1.96) | 0.128 |
| No | 383 (97.21) | 645 (98.93) | 4593 (98.04) | |
| Missing | 105 | 127 | 1158 | |
| History of gestational diabetes (%) | | | | |
| Yes | 39 (9.90) | 64 (9.89) | 490 (10.63) | 0.782 |
| No | 355 (91.10) | 583 (90.11) | 4119 (89.37) | |
| Missing | 105 | 132 | 1234 | |
| History of gestational hypertension (%) | | | | |
| Yes | 39 (9.92) | 98 (15.15) | 646 (14.02) | 0.0476 |
| No | 354 (90.08) | 549 (84.85) | 3963 (85.98) | |
| Missing | 106 | 132 | 1234 | |
| Age at first birth, mean (SD) | 31.31 (4.92) | 29.67 (5.16) | 29.59 (5.28) | <0.0001 |

*Analysis was conducted using ANOVA and $X^2$ tests.
ALSWH, Australian Longitudinal Study on Women's Health; ANOVA, analysis of variance; M-PreM, Menarche-to-PreMenopause.

spectrometry method by the Andrology Laboratory at the ANZAC Research Institute (Sydney, Australia).[31 32] Serum sex hormone-binding globulin, follicle-stimulating hormone and luteinising hormone concentrations were analysed by the Concord Repatriation General Hospital Diagnostic Pathology Unit (Sydney, Australia) using commercial immunoassays (Roche). Anti-Mullerian hormone concentration in serum will be measured by NSW Health Pathology (Sydney, Australia). Total cholesterol, LDL-cholesterol and HDL-cholesterol, and HbA1c were measured by Pathology Queensland (Central Laboratory, Brisbane, Australia) using routine auto-analyser methods. In the future, DNA will be extracted from the stored blood and saliva samples to identify genetic factors associated with reproductive factors, and cardiometabolic and respiratory conditions. The differences in samples should not affect DNA sequences, but could possibly affect epigenetic markers.

### Patient and public involvement
A pilot study was conducted to develop and test the study protocol and provided an opportunity for participant feedback.

### FINDINGS TO DATE
A total of 1278 participants were recruited to the M-PreM Study. Of these, 499 attended a clinical study site and 779 completed a self-assessment. The demographic characteristics of these participants were based on their responses in the eighth survey of the ALSWH (table 3) and compared with the remaining cohort who did not

**Table 4** Clinical outcomes from the Menarche-to-PreMenopause Study

| | Clinic assessment (n=499) | | Self-assessment (n=779) | | |
| --- | --- | --- | --- | --- | --- |
| | n | | n | | P value |
| Age (years), mean (SD) | 499 | 44.60 (1.61) | 779 | 45.28 (1.78) | <0.001 |
| Menopausal status, % | | | | | |
| Surgical menopause | 8 | 1.61 | 20 | 2.73 | 0.531 |
| Not defined due to HRT or OCP use | 177 | 35.61 | 263 | 35.93 | |
| Premenopausal | 202 | 40.64 | 275 | 37.57 | |
| Perimenopausal | 99 | 19.92 | 152 | 20.77 | |
| Postmenopausal | 11 | 2.21 | 22 | 3.01 | |
| Missing | 2 | | 47 | | |
| Body mass index category, % | | | | | |
| Underweight (<18.5 kg/m$^2$) | 2 | 0.40 | 8 | 1.10 | 0.38 |
| Normal weight (18.5–<25 kg/m$^2$) | 202 | 40.56 | 311 | 42.60 | |
| Overweight (25–<30 kg/m$^2$) | 147 | 29.52 | 193 | 26.44 | |
| Obese (≥30 kg/m$^2$) | 147 | 29.52 | 218 | 29.86 | |
| Missing | 1 | | 49 | | |
| Body mass index, mean (SD) | 498 | 27.80 (6.43) | 730 | 27.68 (6.48) | 0.75 |
| Body measurements (cm), mean (SD) | | | | | |
| Waist circumference | 499 | 88.78 (14.84) | 748 | 89.18 (14.98) | 0.647 |
| Hip circumference | 498 | 106.21 (13.49) | 749 | 105.97 (13.62) | 0.755 |
| Arm circumference | 498 | 31.27 (5.13) | 736 | 30.36 (4.35) | 0.001 |
| Thigh circumference | 498 | 59.06 (8.47) | 746 | 55.74 (7.78) | <0.001 |
| Blood pressure (mm Hg), mean (SD) | | | | | |
| Systolic | 494 | 119.11 (12.37) | – | – | n/a |
| Diastolic | 494 | 77.29 (8.84) | – | – | |
| Resting heart rate (beats per min), mean (SD) | 493 | 75.15 (10.50) | – | – | n/a |
| Cardiometabolic biomarkers, mean (SD) | | | | | |
| Total cholesterol (mmol/L) | 489 | 5.21 (0.89) | – | – | n/a |
| HDL cholesterol (mmol/L) | 489 | 1.64 (0.39) | – | – | |
| LDL cholesterol (mmol/L) | 486 | 3.00 (0.77) | – | – | |
| HbA1c (%) | 489 | 5.14 (0.36) | – | – | |
| Handgrip strength (kg), mean (SD) | | | | | n/a |
| Dominant hand | 486 | 30.57 (6.01) | – | – | |
| Non-dominant hand | 486 | 28.62 (5.26) | – | – | |
| Standing balance test (s)*, mean (SD) | | | | | n/a |
| Eyes open | 498 | 28.57 (5.26) | – | – | |
| Eyes closed | 498 | 10.97 (10.36) | – | – | |
| Chair rise test (s), mean (SD) | 495 | 16.72 (4.77) | – | – | n/a |
| Reciprocal of the chair rise test†, mean (SD) | 495 | 6.48 (1.86) | | | |
| Montreal Cognitive Assessment‡, mean (SD) | 499 | 27.69 (1.94) | – | – | n/a |
| Lung function, pre-bronchodilator, mean (SD) | | | | | |
| FEV$_1$ (L) | 283 | 3.03 (0.45) | – | – | n/a |
| FVC (L) | 283 | 3.84 (0.56) | – | – | |
| FEV$_1$/FVC (%) | 283 | 0.79 (0.05) | – | – | |
| Lung function, post-bronchodilator, mean (SD) | | | | | |
| FEV$_1$ (L) | 283 | 3.12 (0.46) | – | – | n/a |
| FVC (L) | 283 | 3.84 (0.56) | – | – | |
| FEV$_1$/FVC (%) | 283 | 0.82 (0.05) | – | – | |

**Table 4** Continued

| | Clinic assessment (n=499) | | Self-assessment (n=779) | | |
|---|---|---|---|---|---|
| | n | | n | | P value |
| Ever had asthma or wheezy breathing (%) | | | | | |
| Yes | 194 | 38.96 | 279 | 36.23 | 0.328 |
| No | 304 | 61.04 | 491 | 63.77 | |
| Missing | 1 | | 9 | | |
| Had asthma or wheezy breathing in the last 12 months (%) | | | | | |
| Yes | 82 | 42.27 | 139 | 49.12 | 0.141 |
| No | 112 | 57.73 | 144 | 50.88 | |
| Missing | 305 | | 496 | | |
| Wheeze or whistling in your chest in the last 12 months (%) | | | | | |
| Yes | 79 | 15.86 | 137 | 17.77 | 0.378 |
| No | 419 | 84.14 | 634 | 82.23 | |
| Missing | 1 | | 8 | | |
| Ever had attacks of cough with phlegm (%) | | | | | |
| Yes | 239 | 51.84 | 361 | 51.87 | 0.994 |
| No | 222 | 48.16 | 335 | 48.13 | |
| Missing | 38 | | 83 | | |
| Ever had cough in the absence of a cold (%) | | | | | |
| Yes | 116 | 25.16 | 141 | 20.17 | 0.045 |
| No | 345 | 74.84 | 558 | 79.83 | |
| Missing | 38 | | 80 | | |
| Ever had phlegm in your chest in the absence of a cold (%) | | | | | |
| Yes | 28 | 6.07 | 49 | 7.01 | 0.531 |
| No | 433 | 93.93 | 650 | 92.99 | |
| Missing | 38 | | 80 | | |

Analysis was conducted using t-tests and $X^2$ tests.
*Maximum score of 30 s.
†Reciprocal of chair rise time×100.
‡Score out of 30.
$FEV_1$, forced expiratory volume in 1 s; FVC, forced vital capacity; HbA1c, glycated haemoglobin; HDL, high-density lipoprotein; HRT, hormone replacement therapy; LDL, low-density lipoprotein; n/a, not applicable; OCP, oral contraceptive.

participate in M-PreM. As expected, those who completed a clinical assessment predominantly lived in a major city, whereas those who participated in the self-assessment were spread among major cities, and inner and outer regional areas. Compared with the participants who completed the self-assessment or did not participate in the M-PreM Study, in-clinic participants were more likely to be university educated and less likely to have three or more children, to ever have smoked or have a history of preterm birth. They also started having children at a later age. Participants in the M-PreM Study, regardless of mode of participation, were more likely to report better health and be non-drinkers than non-participants.

Table 4 shows some of the outcome measurements. At the time of participation, approximately 40% of the women were premenopausal and 20% had begun their menopausal transition, regardless of their mode of participation. Over 55% of participants were categorised as overweight or obese based on their BMI. The BMI and body measurements were similar between the clinic-assessed and self-assessed participants, except arm and thigh circumference, which were larger in clinic participants. Also, the prevalence of respiratory symptoms was similar between the clinic-assessed and self-assessed participants.

### Strengths and limitations
The strength of the M-PreM Study is that it presents an invaluable opportunity to investigate the relationship between reproductive characteristics across the reproductive lifespan and risk indicators for chronic disease and functional and mental health before midlife. In spite of the difficult circumstances due to the COVID-19 pandemic, we have succeeded in recruiting a large sample of participants who have completed questionnaires, clinical measures, physical activity assessments and blood sample collection to augment the existing information available from the ALSWH surveys. The distributions of

measurements were well within the expected ranges found in other studies for handgrip strength,[33] BMI,[34] systolic and diastolic blood pressure,[34 35] cholesterol,[36 37] waist circumference,[34] hip circumference,[38 39] thigh circumference[38 39] and arm circumference.[40] The mean Montreal Cognitive Assessment scores were higher in our cohort compared with other studies,[41 42] possibly due to the high representation of women with university education in our cohort. In addition, data from this study will provide the baseline for a follow-up of participants in 6 years' time to determine the relationship between reproductive history and women's health before and after menopause (figure 2).

The conduct of the study was affected by the coronavirus (COVID-19) pandemic. After two lengthy study suspensions, the participation rate was lower than expected as participants were uncertain about attending hospitals where the study sites were established and people continued to work from home and juggle child care responsibilities, even after the government lifted stay-at-home orders. COVID-19 outbreaks occurred sporadically throughout the study period and study sites were either closed at short notice or their establishment was halted. In addition, the study protocol was modified to minimise risk of COVID-19 transmission between study personnel and participants. Specifically, the objective lung function testing was replaced by a self-reported respiratory health questionnaire. Participation was also affected by the 2019–2020 Australian bushfire disaster, which prevented a study site from being established in Canberra. Another limitation was the location of the study sites. Although ALSWH is a national study, for M-PreM, study sites could only be established in major cities.

Regardless of the limitations, the study will be able to use this wealth of data to make a substantial contribution to understanding female reproductive health.

**Author affiliations**
[1]School of Public Health, The University of Queensland, Brisbane, Queensland, Australia
[2]Allergy and Lung Health Unit, School of Population and Global Health, The University of Melbourne, Carlton, Victoria, Australia
[3]Centre for Women's Health Research, The University of Newcastle, Newcastle, New South Wales, Australia
[4]Institute for Molecular Bioscience, The University of Queensland, Brisbane, Queensland, Australia
[5]Charles Perkins Centre, The University of Sydney, Sydney, New South Wales, Australia
[6]Faculty of Health, Deakin University, Burwood, Victoria, Australia
[7]The George Institute for Global Health, University of New South Wales, Sydney, New South Wales, Australia
[8]Division of Surgery and Interventional Sciences, Faculty of Medical Sciences, University College London, London, UK
[9]School of Women's and Children's Health, University of New South Wales, Sydney, New South Wales, Australia
[10]Gynaecological Research and Clinical Evaluation (GRACE) Unit, Royal Hospital for Women, Sydney, New South Wales, Australia
[11]Medical School, The University of Western Australia, Perth, Western Australia, Australia
[12]Department of Endocrinology and Diabetes, Fiona Stanley Hospital, Perth, Western Australia, Australia
[13]Department of Internal Medicine, Erasmus MC, University Medical Center, Rotterdam, The Netherlands
[14]School of Medicine, The University of Queensland, Brisbane, Queensland, Australia
[15]Mater Research, The University of Queensland, Brisbane, Queensland, Australia

**Acknowledgements** We would like to acknowledge the Australian Government's Department of Health for funding the Australian Longitudinal Study on Women's Health and the women who provided survey data and biomedical data for this study.

**Contributors** GDM is the lead applicant on the funding application and is responsible for the overall content as guarantor. SD, JAV, DL, RRH, AD, GM, HM and JD are co-applicants of the funding application and provided feedback on the design and the operations of the study. SD, JA and BBY were study site principal investigators and provided on-site supervision. ES and MH provided accelerometers and protocols for physical activity data collection. H-WC is the project coordinator for the study and drafted this manuscript. H-FC is the postdoctoral fellow for this study and provided feedback on the study operations. GIM is a postdoctoral fellow for this study and provided statistical expertise. All authors reviewed and approved the final manuscript.

**Funding** This work is supported by the National Health and Medical Research Council (NHMRC) Project Grant (APP1129592). GM is supported by an NHMRC Principal Research Fellowship (APP1121844).

**Competing interests** None declared.

**Patient and public involvement** Patients and/or the public were involved in the design, or conduct, or reporting, or dissemination plans of this research. Refer to the Methods section for further details.

**Patient consent for publication** Not required.

**Ethics approval** This study involves human participants and was approved by the Metro South Health and Health Services Human Research Ethics Committee (reference number: HREC/2019/QMS/52052) and ratified by the University of Newcastle and the University of Queensland Human Research Ethics Committees. All participants provided informed consent by completing an electronic or paper participant consent form.

**Provenance and peer review** Not commissioned; externally peer reviewed.

**Data availability statement** Data are available upon reasonable request. Access to the M-PreM dataset requires approval from the Australian Longitudinal Study on Women's Health (ALSWH) Data Access Committee. More information can be found at the ALSWH website: https://alswh.org.au/for-data-users/.

**ORCID iDs**
Hsiu-Wen Chan http://orcid.org/0000-0003-1545-0488
Jason Abbott http://orcid.org/0000-0002-4406-3121

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
