## [Reviewer comments · BMJ Open]

ARTICLE DETAILS

TITLE (PROVISIONAL)	Cohort profile: A prospective Australian cohort study of women's reproductive characteristics and risk of chronic disease from menarche to premenopause (M-PreM)
AUTHORS	Chan, Hsiu-Wen; Dharmage, Shyamali; Dobson, A; Chung, Hsin-Fang; Loxton, Deborah; Doust, Jenny; Montgomery, Grant; Stamatakis, Emmanuel; Huxley, Rachel; Hamer, Mark; Abbott, Jason; Yeap, Bu; Visser, Jenny; McIntyre, Harold; Mielke, Gregore; Mishra, Gita

VERSION 1 – REVIEW

REVIEWER	Faubion, Stephanie Mayo Clinic Minnesota
REVIEW RETURNED	28-May-2022

GENERAL COMMENTS	This is a report of a cohort profile that will be used in the future to investigate risk of chronic disease based on women's reproductive characteristics based on a subgroup of participants in the Australian Longitudinal Study on Women's Health. It will be exciting to see the data from this prospective cohort study going forward. I have some concerns about the manuscript in its current form. The abstract states the purpose of the M-PreM study, but it does not state the purpose of the current manuscript which is to describe the profile of the cohort. The only place this is clear is in the title. It is hard for the reader to orient themselves to the purpose of the manuscript without this being clearly defined in the abstract and then in the methods. Under Introduction and the first aim, consider changing respirator conditions to respiratory function or respiratory health and delete poor before cognitive (you are assessing cognitive health not poor cognitive health). Similarly in the first sentence after that, change respiratory conditions to respiratory health or function (you are not just assessing for respiratory conditions). Under the second aim, what does body size mean. Is it BMI or waist circumference or both? If both, maybe change to anthropometrics. I'm missing the normal categories of a study, including Methods and Discussion. Need to add Methods as a heading before cohort description. Under exclusion criteria, did you include women receiving chemotherapy for other cancers (not just reproductive)? Chemotherapy for other cancers, e.g., breast, can disrupt ovarian function and hormone measurements. I'm confused as to the overall plan for the study and I think this needs to be clarified in Methods and probably in the Discussion as well. Is the plan to collect the questionnaires and measurements you obtained for this report only once or again in the future? And if the data will be collected again, at what interval? After reading it several times, I think I understand that the ALSWH surveys collected every three years will continue in the future and will be
--

	used as part of this longitudinal study, but again that is not clear. Please clarify for the reader. I think the header Results goes before Findings to Date. Can you clarify who "non-participants" are? are these women in ALSWH who are not in the M-PreM study? If so, please state. There should be a Discussion before Strength and Limitations. In this, you can discuss studies planned or ongoing, any future data collection you anticipate will occur. How long are these women to be followed? A timeline as a figure may help orient the reader on the plan for the longitudinal study and points at which data have been/will be collected. The last paragraph of strengths and limitations is confusing. Need to add "a" before wealth. I don't know what "potentially sufficient number of participants" is. "All major research questions" is broad and an overstatement. Please revise and add some detail here or delete the sentence.
--	---

REVIEWER	Miller, AB University of Toronto Dalla Lana School of Public Health
REVIEW RETURNED	11-Jun-2022

GENERAL COMMENTS	None
------

VERSION 1 – AUTHOR RESPONSE

Reviewer 1: Dr Stephanie Faubion, Mayo Clinic Minnesota

Comment 1: *This is a report of a cohort profile that will be used in the future to investigate risk of chronic disease based on women's reproductive characteristics based on a subgroup of participants in the Australian Longitudinal Study on Women's Health. It will be exciting to see the data from this prospective cohort study going forward. I have some concerns about the manuscript in its current form.*

Response: We appreciate all your comments, and, as you suggested, major changes to the manuscript have been made. These included improvements in the abstract, introduction, and clarifications in the methods. Please refer to the responses to the comments below for further detail.

Comment 2: *The abstract states the purpose of the M-PreM study, but it does not state the purpose of the current manuscript which is to describe the profile of the cohort. The only place this is clear is in the title. It is hard for the reader to orient themselves to the purpose of the manuscript without this being clearly defined in the abstract and then in the methods.*

Response: We have added the following sentences in the abstract and introduction.

Abstract: 'The purpose of this cohort profile paper is to describe the rationale, study design, and participant characteristics of the M-PreM study.'

Introduction: 'This paper describes the recruitment of participants, data collection protocol, and participant characteristics of the M-PreM study.'

Comment 3: *Under Introduction and the first aim, consider changing respirator conditions to respiratory function or respiratory health and delete poor before cognitive (you are assessing cognitive health not poor cognitive health). Similarly in the first sentence after that, change respiratory conditions to respiratory health or function (you are not just assessing for respiratory conditions).*

Response: This has been changed accordingly. The aims and the last paragraph in the introduction now reads as:

Introduction: The Menarche-to-PreMenopause (M-PreM) study aims to: 1) investigate links between reproductive factors and markers of cardiometabolic health, respiratory, and cognitive and functional

health among women before they reach middle-age; and 2) determine whether body size (anthropometry) and physical activity modify these relationships.

The markers of cardiometabolic health measured are glycated haemoglobin (HbA1c), serum lipids, and blood pressure and the markers of respiratory health will be obtained from lung function tests and questionnaires. Other measures include functional and cognitive tests. In addition, this study will use existing data on physical and mental health, reproductive factors and sociodemographic data from the Australian Longitudinal Study on Women's Health surveys and linked administrative health records. This paper describes the recruitment of participants, data collection protocol, and participant characteristics of the M-PreM study.

Comment 4: *Under the second aim, what does body size mean. Is it BMI or waist circumference or both? If both, maybe change to anthropometrics.*

Response: We have changes accordingly. In the second aim, we added '(anthropometry)' after body size and now it links better to Table 2 where all the related measurements are under the subheading of 'Anthropometry'. The aim now reads as 'determine whether body size (anthropometry) and physical activity modify these relationships'. Please refer to "Comment 3" for further updates in the aims and the last paragraph of the introduction.

Comment 5: *I'm missing the normal categories of a study, including Methods and Discussion. Need to add Methods as a heading before cohort description.*

Response: According to the instructions for authors, Cohort Description is the main heading after Introduction, followed by Findings To Date. Therefore, we have kept the current headings as is.

Comment 6: *Under exclusion criteria, did you include women receiving chemotherapy for other cancers (not just reproductive)? Chemotherapy for other cancers, e.g., breast, can disrupt ovarian function and hormone measurements.*

Response: We excluded participants who were undergoing breast cancer. We changed the Exclusion and inclusion criteria section to clarify this, as follows.

Page 6: "Exclusion and inclusion criteria: The 8th survey of the 1973-1978 ALSWH cohort was conducted in 2018 when the participants were aged 40-45 years and completed by 7121 participants. In this survey, women were introduced to the M-PreM study and asked to indicate their interest in participating. Those who responded positively (n=4584) were invited to the M-PreM study (Figure 1). The exclusion criteria included women who were pregnant or had been diagnosed with a breast or reproductive cancer and were undergoing treatment at the time of invitation. Chemotherapy and radiation treatments for breast or reproductive cancers often affect reproductive function and blood sex hormone concentration."

Comment 7: *I'm confused as to the overall plan for the study and I think this needs to be clarified in Methods and probably in the Discussion as well. Is the plan to collect the questionnaires and measurements you obtained for this report only once or again in the future? And if the data will be collected again, at what interval? After reading it several times, I think I understand that the ALSWH surveys collected every three years will continue in the future and will be used as part of this longitudinal study, but again that is not clear. Please clarify for the reader.*

Response: We agree this was unclear. Changes in the Exclusion and inclusion criteria have been made (please refer to "Comment 6" for an updated version of this section). Additionally, in the Findings To Date section (2nd sentence), we modified the sentence to 'Of these, 499 attended a clinical study site and 779 completed a self-assessment for this single time-point data collection.'

Comment 8: *I think the header Results goes before Findings to Date.*

Response: Please see our response to Comment 5.

Comment 9: Can you clarify who "non-participants" are? are these women in ALSWH who are not in the M-PreM study? If so, please state.

Response: Thank you for pointing this out. In the 3rd sentence under Findings To Date, we have removed the words 'non-participants' and changed to the sentence to 'The demographic characteristics of these participants were based on their responses in the 8th survey of the ALSWH (Table 3) and compared with the remaining cohort who did not participate in M-PreM.'

Comment 10: There should be a Discussion before Strength and Limitations. In this, you can discuss studies planned or ongoing, any future data collection you anticipate will occur. How long are these women to be followed? A timeline as a figure may help orient the reader on the plan for the longitudinal study and points at which data have been/will be collected.

Response: According to the instructions to authors, the Findings To Date section is followed by Strength and Limitations so we have not added a Discussion section.

With regards to future plans for the study, we agree that a timeline figure (Figure 2 added) will help illustrate the point that (i) the main ALSWH study is longitudinal and ongoing, and (ii) M-PreM is a substudy with plans of a pre- and post-menopause biomedical data collection.

Under Cohort Description, the 3rd sentence now reads 'Data collection of this currently ongoing study primarily comprises self-report surveys conducted approximately every three years.'

In the Strengths and Limitations section, the first paragraph now finishes with 'In addition, data from this study will provide the baseline for a follow-up of participants in 6 years' time to determine the relationship between reproductive history and women's health before and after menopause (Figure 2).'

Comment 11: The last paragraph of strengths and limitations is confusing. Need to add "a" before wealth. I don't know what "potentially sufficient number of participants" is. "All major research questions" is broad and an overstatement. Please revise and add some detail here or delete the sentence.

Response: We have revised the sentence to 'Regardless of the limitations, the study will be able to use this wealth of data to make a substantial contribution to understanding female reproductive health.'

Reviewer 2: Dr. AB Miller, University of Toronto Dalla Lana School of Public Health

Comments to the Author:

None

VERSION 2 – REVIEW

REVIEWER	Faubion, Stephanie Mayo Clinic Minnesota
REVIEW RETURNED	11-Sep-2022
GENERAL COMMENTS	My concerns have been adequately addressed. No additional concerns.